# Bio-Hacking Better Health—Leveraging Metabolic Biochemistry to Maximise Healthspan

**DOI:** 10.3390/antiox12091749

**Published:** 2023-09-11

**Authors:** Isabella D. Cooper, Yvoni Kyriakidou, Lucy Petagine, Kurtis Edwards, Bradley T. Elliott

**Affiliations:** Ageing Biology and Age-Related Diseases, School of Life Sciences, University of Westminster, 115 New Cavendish Street, London W1W 6UW, UK; y.kyriakidou@westminster.ac.uk (Y.K.); l.petagine@westminster.ac.uk (L.P.); k.edwards2@westminster.ac.uk (K.E.); b.elliott@westminster.ac.uk (B.T.E.)

**Keywords:** ageing, antioxidant, beta-hydroxybutyrate, insulin, hyperinsulinaemia, ketosis, longevity, mitochondria, NAD^+^, ROS

## Abstract

In the pursuit of longevity and healthspan, we are challenged with first overcoming chronic diseases of ageing: cardiovascular disease, hypertension, cancer, dementias, type 2 diabetes mellitus. These are hyperinsulinaemia diseases presented in different tissue types. Hyperinsulinaemia reduces endogenous antioxidants, via increased consumption and reduced synthesis. Hyperinsulinaemia enforces glucose fuelling, consuming 4 NAD^+^ to produce 2 acetyl moieties; beta-oxidation, ketolysis and acetoacetate consume 2, 1 and 0, respectively. This decreases sirtuin, PARPs and oxidative management capacity, leaving reactive oxygen species to diffuse to the cytosol, upregulating aerobic glycolysis, NF-kB and cell division signalling. Also, oxidising cardiolipin, reducing oxidative phosphorylation (OXPHOS) and apoptosis ability; driving a tumourigenic phenotype. Over time, increasing senescent/pathological cell populations occurs, increasing morbidity and mortality. Beta-hydroxybutyrate, an antioxidant, metabolite and signalling molecule, increases synthesis of antioxidants via preserving NAD^+^ availability and enhancing OXPHOS capacity. Fasting and ketogenic diets increase ketogenesis concurrently decreasing insulin secretion and demand; hyperinsulinaemia inhibits ketogenesis. Lifestyles that maintain lower insulin levels decrease antioxidant catabolism, additionally increasing their synthesis, improving oxidative stress management and mitochondrial function and, subsequently, producing healthier cells. This supports tissue and organ health, leading to a better healthspan, the first challenge that must be overcome in the pursuit of youthful longevity.

## 1. Introduction

Throughout history, people have always been fascinated with immortality, and whilst we have seen great increases in human lifespan [1,2,3,4,5], it is not desirable to live longer with chronic diseases. Advances in medical care have enabled increased average lifespan extensions across society; however, this is often accompanied by chronic diseases associated with ageing, such as cardiovascular diseases (CVD), non-alcoholic fatty liver disease (NAFLD), cancer, type 2 diabetes mellitus (T2DM), hypertension, Alzheimer’s disease (AD) and Parkinson’s disease (PD) [6,7,8,9,10,11,12]. The “aim of the game” in ageing and longevity research should be to have a long healthspan with negligible senescence (the absence of biological ageing), such as decreasing measurable functional decline in organs and whole-body fitness, loss of reproductive capabilities and exponentially increasing death rate with chronological age progression [13]. We have successfully extended ageing; our next aim should be to extend youth, and, if we can achieve that, then we may begin to push the envelope on increasing lifespan. If we fail, at least we will have an increased healthspan.

We all instinctively understand what ageing is; however, it is surprisingly difficult to biologically define. We see it manifest from changing appearances, such as loss of skin elasticity and greying hair, to changes in physiology with loss of function, such as decreased fertility in females with each passing decade until complete cessation of menses at menopause. Younger people rarely suffer hip fractures with falls, unlike older people whose bones become weaker where osteoporosis and fragility fracture rates increase [14,15,16]. Moreover, as we age, sarcopenia, the gradual loss of muscle mass, becomes a strong predictor of morbidity and mortality [17,18,19,20,21]. Whilst we are all aware of the macroscopic changes and diseases associated with ageing, it is at the molecular and cellular level where senescence first begins to occur and is associated with corresponding changes in certain biomarkers in plasma (see Figure 1).

Humans are multicellular organisms; our cells operate collectively in biological eusociality. When we consider healthy ageing and extending lifespan, we must first consider context. At the cellular level, the individual cell must not only live longer but must function correctly according to its tissue type, maintaining the differentiated state, limiting replicative cycles and undergoing apoptosis when dysfunction occurs. Unicellular organisms, such as bacteria and yeasts, have a different concept of longevity, having indefinite replication and stasis. Cancer cells behave more like unicellular organisms, they are long-lived and capable of unlimited replicative cycles; they do not function eusocialistically; they de-differentiate, avoid apoptosis and do not go quietly into the night when it has been deemed they are no longer good for the collective whole [22,23]. Our lofty goals as humans would be to maintain optimum cellular and thus organ function, enabling the whole system to thrive, with the aim of ensuring a long healthspan with negligible senescence [24,25] and perhaps a touch of immortality.

Insulin is life-essential and has a hormesis zone. Too much drives the condition hyperinsulinemia, strongly associated with chronic diseases and ageing, as insulin prevents cells from being able to commit controlled cellular “self-destruction” apoptosis when they should [26,27]. Instead, insulin speeds up a cell’s cycle, to replicate itself faster, preventing any pauses to check if DNA copying was conducted properly, and preventing a good dose of intracellular housekeeping [13]. Instead, insulin signals to the cell that energy is abundant, and, therefore, there is no need to be conservative with resources and run a tight ship. Thus, although life essential, we may view excess insulin as an ageing hormone, one that prevents our ability to produce our own powerful anti-ageing ketone body D-beta-hydroxybutyrate (BHB) [28,29,30,31].

Deterioration in our bodily systems as we age, such as chronic insulin signalling, is mechanistically associated with increased incidence of cognitive decline, such as Alzheimer’s disease, malignancies, CVD, deterioration of skin elasticity, decreased exercise performance and loss of muscle strength [30].

## 2. Cellular Ageing

The “damage/error” theory of ageing at the cellular level may be summarised as a cell’s rate of damage versus its rate of repair [32]. Where over time, rate of repair cannot at least match the rate of damage leading to cellular dysfunction and ageing [13,19]. The accumulation of damage within a cell ultimately begins to manifest as cellular dysregulation, where the cell no longer “behaves correctly” as part of a collective of cells that make up the tissues of the organ in which the cell is located. This dissociation from regulated cellular behaviour and/or de-differentiation is the modus operandi of the cancer cell phenotype [23]. The ageing cell also begins to send out destructive inflammatory signals, such as from cytokines and reactive oxygen species (ROS), that signal neighbouring and distant cells in other organs, eliciting adaptive and, often when chronic, maladaptive homeostatic responses to these biological messengers [19]. This ultimately results in systemic changes in organ and tissue function which precipitate as macroscopic age-associated changes to the body which we are all familiar with.

## 3. Oxidative Stress Management; Reactive Oxygen Species Are Intimately Tied to Healthspan

In a healthy individual, both young and old, wayward cells implement active cellular response management systems of repair and/or removal and replacement, these include the following: autophagy, mitophagy and mitochondrial biogenesis, the “eating up, breaking down and recycling” of intracellular organelles, that have been registered as damaged or impaired [33,34,35,36,37,38]. Thus, these processes are essential for containing and preventing the accumulation of intracellular damage, reducing dysregulated cellular behaviour and regulating the production and release of inflammatory signalling molecules.

Apoptosis, controlled cell death, is another physiological cellular mechanism that enables the well-regulated removal of the whole cell, when the damage is beyond repair. Apoptosis is greatly diminished in individuals with chronic diseases and absent in cancer cells [34,39]. Loss of apoptotic function results in long-lived dysfunctional (senescent) cells which promote inflammation and further dysregulation both locally and in distant tissues.

These intracellular housekeeping processes enable the culling of inefficient and even potentially toxic cells from the herd that makes up a tissue, and thus the upkeep of a healthy, optimally functioning organ [40]. However, over time, a cell’s ability to trigger autophagy and/or apoptosis may become impaired for a variety of reasons. Perhaps the stimulus driving apoptosis is not strong enough, or a counter pathological stimulus prevents apoptosis and the gradual dysfunction within the cell sneaks by, undetected and able to keep causing destructive mischief. Over time, accumulation of these cells begins to count as a substantial population in an organ or tissues; at this point, we begin to see manifestations at macroscopic levels with overt pathogenesis and chronic disease symptoms [41].

This internal quality control is governed by a multitude of factors, such as the availability and utilisation of the type of cellular fuelling substrate that impacts intracellular cytosolic nicotine adenine dinucleotide (NAD^+^) availability. NAD^+^ is a key antioxidant, reduced availability is known to profoundly affect mitochondrial (mt) ROS production and management, and epigenetic gene regulation [38,42,43,44]. In the generation of two acetyl-moieties, glucose fuelling consumes a greater amount of NAD^+^ (four) than fatty acid, BHB and acetoacetate combined, which consume two, one and zero NAD^+^, respectively. Therefore, cellular fuelling substrate type has consequences beyond energy provision, as many cellular repair enzymes are NAD^+^-dependent, such as sirtuin 1 and 3 (SIRT1, SIRT3), and the upregulation in transcription of antioxidants such as manganese superoxide dismutase (MnSOD2) and glutathione (GSH) [45,46,47,48,49]. Thus, the type of fuelling substrate cells use to power their energy-dependent activities, instigate different intracellular signals and phenotype adaptation. Furthermore, nutrient-sensing hormones like insulin not only aid in the uptake of glucose, insulin also inhibits beta-oxidation, ketogenesis and ketolysis, thereby enforcing glucose fuelling as a primary substrate [50,51,52], consequently playing an integral role in intracellular housekeeping functions and capacity.

Energy is required for the work cells must perform to maintain order in the fight against ever increasing entropic disorder. The intracellular organelles, mitochondria, are charged with producing the largest bulk of energy for the majority of our cells [53]. These dynamic organelles are both a source of life-sustaining energy and a source of destruction, from the processes of capturing chemical energy, via the breakdown of nutrients with the use of oxygen, into the highly efficient small molecule adenosine-triphosphate (ATP) [33,41,54,55].

Oxygen, in turn, is both life-giving and corrosive, having a Goldilocks zone termed hormesis, in which too much (oxygen) is just as dangerous as not enough [53,56]. Free radicals are produced during oxidative phosphorylation (OXPHOS), from oxygen in the process of ATP production via the inner mitochondrial membrane (IMM) electron transport chain (ETC), situated on the matrix side of the cristae [57,58]. A class of free radicals produced include the following: superoxide (O_2_^•−^), an oxygen molecule with an extra electron; hydroxyl (-OH) or hydrogen peroxide (H_2_O_2_), the same found in household drain cleaner, albeit at a different concentration. Collectively, these are termed ROS [33,41].

It was postulated in the 1950s that ROS toxicity is a key player in the ageing process [17]. As research has progressed in elucidating mechanisms of ageing and chronic diseases that limit lifespan, substantial amounts of evidence have accumulated to support this theory. This includes showing an inverse relationship between mtROS production and lifespan, where greater amounts of mtROS generation results in decreased healthspan, as evidenced in chronic diseases that considerably reduce life expectancy [19,59].

## 4. Mitochondrial Hormesis

In addition to cellular energy production, mitochondria are involved with a vast number of cellular processes, including the regulation of metabolic flux [50]. They are charged with the job of performing anabolic processes to furnish cells with precursors for lipid, amino acid and nucleotide biosynthesis and are key regulators of apoptosis [60]. Mitochondria constitute a significant proportion of our cells, increasing in numbers in cells that are highly metabolically active [61,62,63]. The cells of the brain consume 20% of the body’s resting metabolism predominantly via mitochondrial ATP production, yet the brain constitutes 2% of body mass [64], and it is estimated that a single neuron of the substantia nigra may contain up to 2 million mitochondria [62].

Mitochondria are the greatest source of intracellular free radical ROS production, in close proximity to mitochondrial DNA (mtDNA) and the IMM containing the ETC protein complexes. Mitochondria also largely detoxify ROS within themselves [65], and, whilst ROS are considered damaging to a cell’s internal structures and components, they are also signalling molecules in their own right. Ironically, some amount of ROS is necessary for health and life-span extension, that is the element of hormesis, the Goldilocks zone [66,67]. ROS act like a form of SOS or Morse code signalling system, enabling mitochondria to communicate with the cell’s nucleus, a mitochondrial–nuclear cross-talk, otherwise known as mtROS-retrograde signalling [40,53,56,66].

This begs the question, like the chicken and egg conundrum, what is driving cellular behaviour? The genes in the nucleus, or signals from outside and within the cell from the mitochondria, transmitted to the nucleus to activate/elicit adaptive responses, altering gene expression as an epi-phenomena and consequently altering cellular outcomes [23,40,68,69]. Some of those responses lead to the increase in the production of antioxidant enzymes, such as MnSOD2 and catalase, to counter mtROS and maintain the differentiated state [46,70]. Vitamin C, having antioxidant properties, is an essential vitamin that humans must consume; however, humans also possess the ability to synthesise powerful endogenous antioxidants. These endogenously derived antioxidants, such as reduced GSH and glutathione peroxidase (GPx), are able to recycle vitamin C [49]. Just the right amount of ROS drives a cell to make more “fresh and healthy” mitochondria, via mitochondrial biogenesis, as ROS damages the ones in existence, thus helping to decrease ROS production once again. The physiological functions of ROS free radicals that are needed for cell signalling to induce health-promoting adaptations, such as for athletes, may be hampered by excessive antioxidant supplementation at key moments during or post exercise-induced ROS production [71].

## 5. ROS on OXPHOS via Cardiolipin Upregulate Aerobic Glycolysis/Pseudo-Hypoxia-Signalling Cascades, a Cancer Phenotype

Chronic excessive ROS leads to greater oxidative damage than rate of repair, such as lipid peroxidation of ROS-sensitive phospholipid cardiolipin (CL). CL is biosynthesised by the mitochondria and is almost exclusively found in the cristae of the IMM. CL plays an essential role in the structural integrity of supporting respiratory chain supercomplexes, stabilising tertiary and quaternary ETC complexes, substrate carrier proteins, mitochondrial permeability transition pores and ATP-synthase. CL is an integral component in energy transducing membranes, whilst its unique dimeric-cross-linked phospholipid structure and inherent anisotropy enables the high curvature formation found in cristae. CL is required for the trans-membrane protein-phospholipid complex IV cytochrome C oxidase (CcO). CcO is composed of 13 subunits; subunits I, II and III form the catalytic core of the enzyme and are encoded by mtDNA. CcO functionally requires CL, stabilising its structural integrity, a necessary requirement for the transport of electrons and translocation of protons [57]. Electron transport activity is diminished by 50% with the removal of CL, due to the dissociation of the respiratory chain complexes [72]. CL stabilises and acts as an allosteric ligand for succinate dehydrogenase (SDH) and is essential for the curtailment of ROS production at this location.

CL is tightly bound to complex V, where it is found to be critical in oligomerisation and ATP-synthase assembly [57,58]. Depletion of CL correlates to abnormal mitochondrial morphology and subsequent OXPHOS capabilities as well as increased ROS production [57]. Cells are unable to respire effectively when they have an oxidised CL composition or a decreased content of CL. Furthermore, no tumour cells have yet to be shown to have normal content and composition of CL [73]. Oxidised CL results in increased ROS, positively feeding forward, producing further oxidative damage. ROS diffuse to the cytosol, inhibiting hypoxia inducible factor (HIF) prolyl hydroxylase (PHD) enzymes, resulting in the stabilising of HIF-1α, which is then able to interact with HIF-ß, activating the hypoxia-signalling pathway, even when oxygen is present.

Increasing protein stability of HIF-1α leads to upregulated hypoxia signalling, ROS directly mediate this by oxidation of HIF-1α amino acid residue Cys533 [74]. Furthermore, indirect upregulation of HIF-1α by ROS is mediated via downregulation of the deacetylase sirtuin enzyme, SIRT1, resulting in the maintenance of HIF-1α amino acid residue Lys647 acetylation, aiding and abetting in the stabilising of HIF-1α [75]. HIF-1α- HIF-ß interaction consequently activates inducible HIF-transcription factors (HIF-TF) to initiate pseudo-hypoxia signalling cascades, resulting in the upregulation of aerobic fermentation, a cancer cell phenotype often referred to as the Warburg effect [76,77,78]. Further effects are in the modulation of gene expression to activate hypoxia mediated responses, such as enhanced gene expression for increased angiogenesis, erythropoiesis, cell cycle and survival mechanisms [79]. Upregulated HIF signalling increases glucose transporter GLUT1, enabling increased glucose uptake and phosphorylation, further consolidating the upregulation of aerobic fermentation and catabolism of the NAD^+^ pool, leading to a reduction in endogenous mtROS neutralising antioxidants, which then result in more oxidative damage to CL, over chronic timeframes driving a vicious feedforward cycle [80].

These ROS-induced processes culminate in a dramatic increase in glycolysis and glucose shunting to the pentose phosphate pathway (PPP). HIF activation transactivates the gene encoding the enzyme pyruvate dehydrogenase kinase-1 (PDK1) [81], which in turn inhibits pyruvate dehydrogenase. This inhibits the oxidation of pyruvate to acetyl-Co-enzyme-A (ACoA) in the mitochondria, whilst activating lactic acid dehydrogenase, resulting in increased aerobic fermentation of glucose [60]. ROS signal-driven aerobic fermentation of glucose to lactate acid is a metabolic hallmark phenotype of cancer. Furthermore, mitochondrial ROS production may also be induced by hypoxia, activating ROS signalling pathways to promote adaptive transcription programs [79].

Hypoxia is a nearly universal and distinguishing hallmark of cancer growth [82]. The epigenome and transcriptome are sensitive to the cellular metabolic state, chronic hypoxia, coupled with increased ROS instigate inflammation and glycolytic enzyme upregulation signalling pathways, leading to nuclear gene expression of the tumour phenotype [42,43]. Perpetuated over time, this causes irreversible nuclear and mitochondrial genomic damage and oncogenic transformation, a downstream epiphenomenon of impaired cellular respiration [83,84].

CL is an essential phospholipid required for the catalytic activity of the ETC enzymes [57]. Increased lipid peroxidation results in changes in CL fatty acid profile and a total concentration of CL within the IMM. As a result, ETC efficiency reduces, increasing ROS production, concomitantly reducing OXPHOS ATP-production capacity. CL is required for the maintenance of the highly invaginated mt cristae and ATP synthasome assembly. The superoxide O_2_^•−^ anion is most frequently generated at the ETC complex I NADH CoQ oxidoreductase when the mitochondrial matrix NADH/NAD^+^ ratio is high and especially when CL is oxidised by ROS. ROS induced CL peroxidation induces a vicious positive feed forward cycle [58]. If sustained chronically, this positive compounding cycle results in the accumulation of pathological mitochondria that dominate the hierarchy of intracellular signalling. This is seen in the pathogenesis of cellular tumourigenic transformation, with the upregulation of aerobic glycolysis via ROS mediated direct mechanisms, such as stabilising HIF-1α and upregulating NOD-like receptor family pyrin domain-containing 3 (NLRP3) inflammasome assembly, or indirectly via histone deacetylase activity altering gene expression and transcription, leading to substantial changes in cellular phenotype. Consequently, these chronic pathological signals dictate a direction towards greater entropic disorder resulting in cellular, tissue and organ system functional decline and disease [41,46,70].

## 6. Excess ROS Alters the Mitochondrial Structure, Which Dictates Function

The structure of mitochondria is intimately linked to their function; the ultrastructure of tumour mitochondria are often found to be significantly different from the ultrastructure of normal cell mitochondria [85]. Tumour cells tend to have fewer mitochondria, and many have structurally abnormal morphology, with less or completely absent cristae, the site of OXPHOS. This would promote cellular need to upregulate glycolysis and glucose consumption in order to satisfy cellular energy requirement and for the PPP NADP^+^-dependent reduction of oxidised glutathione (GSSH) to reduced GSH, to counter elevated ROS to enable tumorous cells to survive their increasingly acidic intracellular environment. The degree of decreased mitochondrial numbers and abnormal morphology was shown to be directly correlated with breast tumour degree of malignancy [86].

Mitochondrial ROS damage without commensurate counter ROS management can increase cellular need for glucose due to impaired and diminished OXPHOS capacity. Excess and enforced glucose fuelling then depletes cytosolic NAD^+^, further impairing ROS management programs directed through NAD^+^ dependent SIRT1 and SIRT3 [87,88,89]. Furthermore, OXPHOS-driven ATP synthesis is reduced when cellular need for NAD^+^ replenishment demand is high, thus further driving aerobic glycolysis activity and demand [44,47]. Through multiple compensatory mechanisms that alter nuclear gene expression, chronic elevated ROS may ultimately result in permanent change in mitochondrial superstructure and function, thus a heavier dependency on glycolysis and altered intracellular epigenetic modulation. Concurrently, extracellular stimuli may also drive metabolic transformation. Metabolic syndrome (MetS) and T2DM are typified by a state of hyperinsulinemia. Hyperinsulinemia is considered a risk factor for cancer [90]. Hyperinsulinemia, hyperglycaemia and inflammation are suffered by type 2 diabetics; this may be a link as to why diabetics have an increased risk of cancer [91].

Insulin increases mitochondrial ceramide synthesis, directly increasing mitochondrial ROS [92], leading to increased fission protein Drp1 relative to fusion protein Mfn2 [93]. This results in increased mitochondrial fission leading to impaired mitochondrial energy production and cellular pathology, as seen in osteoporosis, cancer, beta-cell failure, and neurodegenerative diseases including Alzheimer’s and Parkinson’s diseases. Clearly, healthy mitochondria, indicated by structure and function, are required for optimal cellular function. The biochemical processes which occur in ageing as a result of hyperinsulinemia are described in Figure 2.

## 7. Relationship between Calorie Restriction, Beta-Hydroxybutyrate, Low-Insulin and Longevity

Calorie restriction studies across many species, from the single celled yeast to the nematode worm *Caenorhabditis elegans*, to mice through to primates, have been shown to extend lifespan with healthspan [46,49,56,70,94]. The mechanism of how this is achieved has not been fully resolved; however, a number of key aspects of calorie restriction define the metabolic changes that are induced within the organism and ourselves included, namely ketosis [49]. Endogenous ketone synthesis is termed ketogenesis. This is the production of the small molecule BHB, a ketone body that humans synthesise when fasting and/or in a low insulin secretion state. Ketones are not only a source of fuel for most cells in the body, they also act as signalling molecules, effectively telling the intracellular organelles, the mitochondria and nuclei, what is going on outside of the cell and organism, and how best to adapt and respond [46,70,95]. Nutritional ketosis (NK) is hallmarked by BHB concentrations of ≥0.5 mmol/L in the bloodstream [16,89]. Measurable blood BHB with normoglycaemia has been shown to hold a hint of an elixir to a healthier if not longer life [53].

## 8. Cellular Energy Status Sensing

It is essential for cells to “know” what the energy status of the whole organism is and thus also for itself, as building more cellular components in preparation for cell division would be wasteful if there were not enough nutrients available. The energy and nutrient status signalling metabolite BHB is synthesised largely by hepatocytes (cells that also perform ketogenesis include enterocytes, renal tubular epithelial cells, astrocytes, retinal pigment epithelium cells and myeloid cells) from fatty acids derived either from adipocytes or from fat within a meal, when insulin is low, thus no longer inhibiting ketogenesis [49,54,70,96,97]. BHB is released into the bloodstream to be used in extrahepatic tissues to generate the necessary ATP required for a cell’s survival and function when glucose is not provided exogenously [98]. BHB is produced as a result of fasting or restricted energy provision to Animalia organisms and/or when insulin secretion levels are below a ketogenesis-inhibiting threshold [99]. When BHB levels are elevated, it acts as a systemic signalling molecule from hepatocytes, altering cellular gene expression and behaviour to adapt to the signal that is telling them, “The organism is not receiving much food/nutrients if any at all” and is utilising internal energy stores from fatty acids converted to BHB via ketogenesis. BHB signalling co-ordinates a systemic shift in a cellular system’s phenotype towards conservation (slowing down the cell cycle/decreasing mitotic rate) and increasing recycling of cellular materials (autophagy and mitophagy), culminating in greater cellular health and survival in the face of starvation [70,99].

BHB and NAD^+^ increase SIRT1 activity, rescuing mitochondrial function and DNA repair via mono- or poly-ADP-ribosylation polymerase enzymes (PARPS) in premature ageing progressive neurodegeneration [100,101]. Chronic hyperinsulinaemia, as detected by its effect on suppressing ketogenesis, depletes NAD^+^ and BHB [47]. BHB activates the HCAR2 receptor, decreasing PI3K activity, resulting in decreased mTOR activity [46,102]. mTOR is an intracellular nutrient and growth factor sensor and mediator of cellular adaptive responses. Decreased mTOR activity increases autophagy and mitophagy, preserving protein status, and thus decreasing muscle loss in times of fasting [52]. BHB interacts with the inflammasome in immune cells to reduce inflammatory cytokine production [103].

If during times of fasting muscle cells continued to utilise glucose for fuel instead of fatty acids, this would decrease glucose availability for mature red blood cells (RBC) and the brain. Thus, hepatocyte BHB signalling reduces myocyte glucose uptake whilst increasing transcription of mitochondrial OXPHOS protein complex genes in order to increase beta-oxidation capacity. Concomitantly, this increases gene transcription for antioxidative enzymes, improving ROS management via histone deacetylase inhibition, upregulation of NAD^+^ sirtuin activity and directly modulating chromatin via beta-hydroxybutyrylation, resulting in decreased basal ROS levels, increasing cellular resilience and survival capacity [46,70,102]. Ultimately coordinating cellular function in a multicellular organism into an integrated physiological system.

## 9. Hyperinsulinaemia and Insulin Resistance

Hyperinsulinemia is often both a driver and result of insulin resistance, the two are tethered to one another [104]. It is arguable whether insulin resistance precedes hyperinsulinemia; insulin resistance is defined as a higher than normative population concentration of insulin to maintain euglycaemia [105]. However, hyperinsulinaemia may be higher than a healthy level of insulin for an individual but still within the population reference range, and therefore putting the individual at risk of pathology and subsequent late detection, delaying earlier intervention [16,106]. Hyperinsulinemia is technically not just referring to the loss in glycaemia management as seen in “insulin resistance”, as insulin resistance is the term given when viewing insulin in the context of glucose regulation. The insulin resistant state is determined by a lack/reduced level of response to insulin-stimulated glucose uptake and, in this context, the other insulin-signalling pathways, namely the growth and division signal transduction MAPK and PI3K pathways are upregulated. This means that cells that do not take up glucose in insulin resistance, are still affected by the insulin signalling via its other pathways—PI3K and MAPK. Therefore, these glucose uptake insulin resistant cells are not actually 100% insulin resistant as if there were no insulin present [107]. Hyperinsulinemia may or may not maintain glucose homeostasis and is a state of insulin level that starts to drive pathological changes, first at the sub-cellular level and later at a macroscopic tissue and organ level. The very first stage of hyperinsulinemia is when insulin is likely within normative population reference ranges with glucose and HbA1c also being normal, and therefore is not diagnosed as insulin resistance. Then, the question would be how do we know if insulin is at a level (hyperinsulinaemia spectrum) that is likely to be less healthy over time? If we agree that BHB with normoglycaemia in longevity and ageing research is healthy and associated with better healthspan and lifespan outcomes, then a person’s individual insulin level should be at a concentration that ketogenesis is not inhibited whilst maintaining euglycaemia and a healthy HbA1c. We postulate that a level of insulin where euglycaemia is maintained and ketosis is not inhibited, evidenced by detectable BHB levels above 0.3 mmol/L, is a heathy insulin level. Correspondingly, an insulin level that maintains euglycemia whilst inhibiting ketogenesis begins to shift into the early stages of hyperinsulinaemia, a state preceding hyperglycaemia by up to 24 years [27,89]. An insulin level that suppresses ketogenesis for a chronic amount of time to the point that an individual wakes up with less than 0.1 mmol/L and less than 0.3 mmol/L before dinner time of BHB on a capillary ketone meter many days in a row, and does not recover any ketones above 0.1 mmol/L four hours into an OGTT, indicates the insulin levels are suppressing beta oxidation, increasing glycogenolysis, suppressing ketogenesis and by default in doing that, depleting cytosolic NAD+ and reducing cellular antioxidant capacity and net-net increasing oxidative harm on their mitochondria.

## 10. Applications of Biochemistry into Behaviours

Whilst caloric restriction models have been successful in animal models for which nutritional intake can be carefully controlled, such interventions in humans are rarely successful [108]. As an alternative, fasting mimicking diets, such as eating in a narrow time frame within a 24-h window or very low carbohydrate, high healthy fat with moderate protein diets (LCHF) [109], known as ketogenic diets (KD), are also able to induce ketosis in individuals, without the conscious effort of calorie restriction [102,110,111]. The result is decreased dietary induced stimulus of insulin secretion from the pancreas, due to decreased glucose nutrient sensing, the primary driver of insulin secretion, which is predominantly due to dietary starchy carbohydrates found in bread, pasta, rice, flour, corn and sugar. What results is lower concentrations of insulin in the bloodstream, lower blood glucose and increased ketone body BHB. If sustained over chronic time frames, this induces a series of adaptive changes within cells, to shift their intracellular machinery to utilise fat and ketones instead of glucose for fuel. However, this is only the tip of the iceberg in changes induced by the presence of ketones, as well as the decrease in the insulin signalling effect on cells [53,112].

The combination of increased signalling from ketones, coupled with decreased signalling from the growth hormone insulin, propagates intracellular adaptive responses that result in increased efficiency in ATP production with increased intracellular housekeeping activities. This leads to cells being able to remove and replace old organelles (autophagy and mitophagy), ensuring a decreased residual damage load that gradually impedes a cell’s function. This also translates into more time for DNA to be checked by DNA housekeeping proteins, of which there are also more as a result of being in an endogenously induced state of ketosis, metabolic phenotype 1 [106]. This prevents DNA duplication errors from being propagated into the next cell division’s daughter cells, increasing genome stability and, thus, reducing the risk of potential replicative errors, cancers or other age-associated diseases [49,54,70].

## 11. Conclusions

Insulin in some regard is the nemesis of ketones, given insulin suppresses our ability to produce ketones and subsequently deprives us of the anti-ageing properties associated with this small molecule [54]. Decreased insulin signalling has been shown to increase healthspan and lifespan [46,70]. Although, once again, insulin is life essential, too much drives hyperinsulinemia, preventing well-regulated apoptosis [26,27]. Instead, insulin increases cell cycle and replication rate, reducing the inspection of DNA copying quality and intracellular housekeeping [13]. Insulin signals to the cell, energy abundance, increase growth and division, whilst decreasing repair. Excess insulin may be viewed as an ageing hormone, preventing self-production of the powerful anti-ageing ketone body BHB [28,29,30,31]. Chronic insulin signalling, hyperinsulinaemia, is mechanistically associated with increased incidence of chronic diseases of ageing, including the following: CVD, cancer, cognitive decline and dementias such as Alzheimer’s and Parkinson’s diseases, and T2DM [30]. Many of these ageing-associated conditions have been shown to be decreased by the production and use of ketone bodies, a normal and healthy metabolite that we are able to produce when we do not overstimulate the production of insulin through our dietary choices [49,113]. Sustaining NK levels of BHB enables the prevention of vicious insulin-signalling effects on the quality of healthspan and ageing. BHB is able to directly neutralise ROS, thus making it a powerful antioxidant that we can make ourselves [53,114]. We are often told to eat to keep up our energy and health; however, perhaps a little less, results in a little more with regards to healthspan and lifespan. Instead of having to achieve this through calorie restriction, we can hack our bodies’ energy signalling system, by either eating only once a day, but as much as one wants, or eating foods that do not stimulate much insulin production. The result is the same as fasting and calorie restriction, one has less insulin and more ketones, which in turn translates into healthier cells, a healthier individual, and a chance to maximise lifespan potential.

## Figures and Tables

**Figure 1 antioxidants-12-01749-f001:**
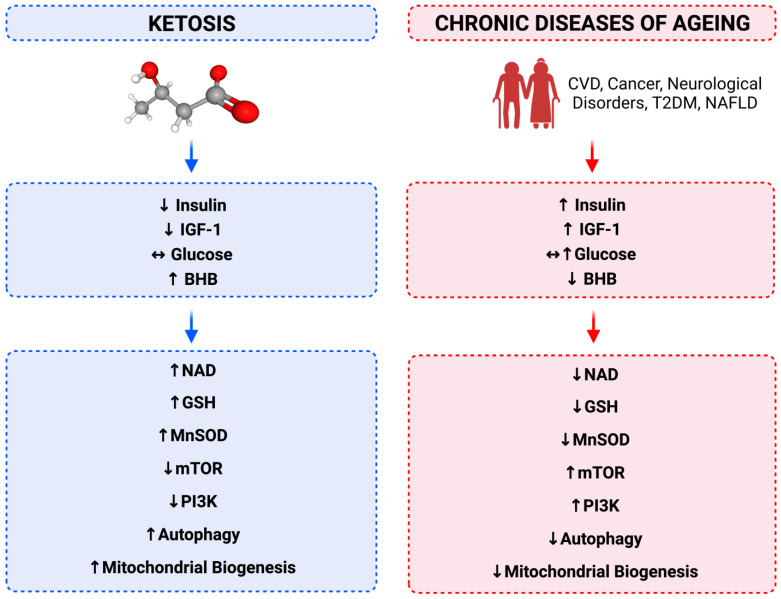
A comparison between the effects of ketosis (blue boxes) versus commonly and chronically not being in ketosis (red boxes) on biomarkers and intracellular markers associated with healthspan and longevity versus chronic diseases associated with ageing. Ageing is a physiological process underpinned by many cellular and molecular mechanisms. Metabolic hormones and metabolites strongly dictate cellular health trajectory and play a crucial role in the development of age-related diseases. Common age-related diseases include cardiovascular diseases (CVD), non-alcoholic fatty liver disease (NAFLD), cancer, type 2 diabetes mellitus (T2DM), hypertension and neurodegenerative disease. Beta-hydroxybutyrate (BHB), glutathione (GSH), insulin-like growth factor 1 (IGF-1), manganese superoxide dismutase (MnSOD), mechanistic target of rapamycin (mTOR), nicotinamide adenine dinucleotide (NAD), phosphatidylinositol 3-kinase (PI3K).

**Figure 2 antioxidants-12-01749-f002:**
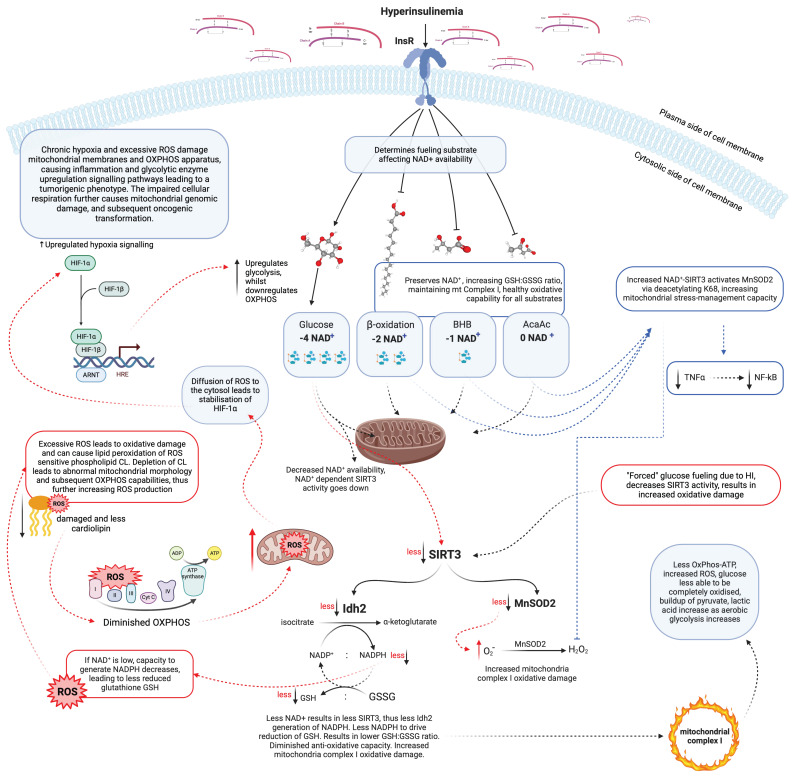
Schematic representation of hyperinsulinemia signalling on intracellular antioxidant regulation dynamics, oxidative damage and mitochondria. Hyperinsulinaemia enforces glucose fuelling, leading to increased hypoxia-signalling pathways and decreased antioxidant expression and activity. This leads to increased reactive oxygen species (ROS) production, leading to increased oxidative stress, damaging oxidative phosphorylation (OXPHOS) proteins and cardiolipin (CL), further increasing inflammatory-signalling pathways, altering mitochondrial morphology leading to dysfunctional mitochondria which signal to upregulate aerobic fermentation, a tumourigenic phenotype. Acetoacetate (AcAc), aryl hydrocarbon receptor nuclear translocators (ARNT), adenosine triphosphate (ATP), beta-hydroxybutyrate (BHB), cardiolipin (CL), hyperinsulinaemia (HI), hypoxia Inducible Factor 1 Subunit α (HIF-1α), hypoxia Inducible factor 1 Subunit β (HIF-1β), hypoxia response element (HRE), insulin receptor (InsR), isocitrate dehydrogenase 2 (Idh2), glutathione oxidised form (GSSG), glutathione reduced form (GSH), nicotinamide adenine dinucleotide (NAD^+^), nicotinamide adenine dinucleotide phosphate (NADP^+^), nuclear factor-kB (NF-kB), oxidative phosphorylation (OXPHOS), reactive oxygen species (ROS), sirtuin 3 (SIRT3), superoxide (O_2_^•−^), tumour necrosis factor α (TNFα).

## Data Availability

No new data were created or analyzed in this study. Data sharing is not applicable to this article.

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
