# Peer review of "Bio-Hacking Better Health—Leveraging Metabolic Biochemistry to Maximise Healthspan"

_antioxidants, 2023, doi:10.3390/antiox12091749_

Round 1
Reviewer 1 Report
his article reviews the significance of antioxidants and ketosis for preventing ageing and senescence to push the envelope on increasing our lifespan.
The comments are;
1. Figure 1 does not show well the implication for this article. Especially, the relationship between “Ketosis” and “Chronic diseases of ageing” is not clear.
2. Section 2 should be combined with “Introduction”.
3. It would rather be good that Sections 4, 5, and 6 are combined together.
4. The titles of Section 8, 9, and 10 might not suit well for each content. Check them carefully.
5. The position of Figure 2 is not adequate. It would rather be moved into the section 9.
6. The description of L413-425 should be moved into the “Introduction”.
7. The authors could note a significance of BHB for a resistance to vicious insulin signaling during ageing in the “Conclusions”.
Author Response
Dear Reviewer,
Thank you very much for your detailed and valuable feedback. We have taken on board all of your beneficial remarks and are pleased to have addressed them all. Please see the in-text revisions shown in blue colour in the draft manuscript, and notes below:
Point 1. Figure 1 does not show well the implication for this article. Especially, the relationship between “Ketosis” and “Chronic diseases of ageing” is not clear.
Response 1: Thank you for your comments, which was useful, as we realise that there was a need to improve the figure legend in explaining this too. Please see lines 55 and 58-59, in the manuscript in blue. Figure 1 legend explains that this figure is a listed comparison on biomarkers and intracellular markers, between ketosis and chronic diseases of ageing.
Point 2. Section 2 should be combined with “Introduction”.
Response 2: Thank you for your advice, we have combined section 2 into the introduction. Please see line 67.
Point 3. It would rather be good that Sections 4, 5, and 6 are combined together.
Response 3: Thank you for your advice, we have combined sections 4, 5 and 6 together. Please now note that due to combining section 2 with the introduction, section 4 has now become section 3 and have given this a new section title. Please see lines 110-172.
Point 4. The titles of Section 8, 9, and 10 might not suit well for each content. Check them carefully.
Response 4: Thank you for your recommendation, your advice was very helpful and has helped us to improve these titles. We have amended them, and they now better reflect their corresponding paragraphs. Please see lines: 211-212, (formerly section 8, reclassed as section 5), 289 (formerly section 9, reclassed as section 6), 323 (formerly section 10, reclassed as section 7). Please note that section reclassification is due to the combining of sections.
Point 5. The position of Figure 2 is not adequate. It would rather be moved into the section 9.
Response 5: Thank you for your feedback. We have taken your advice and have moved the position of the in-text reference into section 9 (now section 6) indicated on lines 320-321, with the figure following after on Page 8 of 18, post section 6 (formerly section 9).
Point 6. The description of L413-425 should be moved into the “Introduction”.
Response 6: Thank you for your recommendation which makes sense and improves the introduction. We have moved this section of text to lines 80-93, and have amended the conclusions, please see lines 424-432.
Point 7. The authors could note a significance of BHB for a resistance to vicious insulin signaling during ageing in the “Conclusions”.
Response 7: Thank you for this suggestion, we have incorporated this into the conclusion in lines 460-477.

Reviewer 2 Report
The hormesis-related aspect of insulin in aging, by regulating the production of ketones and antioxidant systems, is explored in this perspective article. It is an interesting discussion, but how does the insulin resistance fit in this discussion? Hyperinsulinemia rises due to insulin resistance, which is a typical condition in aged organisms. Thus, even with high insulin concentrations, insulin signaling is impaired in different tissues.
Mitochondrial fission and fusion are not specifically discussed in item 9, as the title suggests.
Author Response
Dear Reviewer,
Thank you very much for your detailed and valuable feedback. We have taken on board all of your beneficial remarks and are pleased to have addressed them all. Please see the in-text revisions shown in blue colour in the draft manuscript, and notes below:
Point 1. How does the insulin resistance fit in this discussion? Hyperinsulinemia rises due to insulin resistance, which is a typical condition in aged organisms. Thus, even with high insulin concentrations insulin signalling is impaired in different tissues.
Response 1: Thank you for your comments, which was taken on board. We have added a whole secEon on this topic which we are glad to have been made aware of as it certainly improves the manuscript. Please see lines: 392-428.
Point 2. Mitochondrial fission and fusion are not specifically discussed in item 9, as the title suggests.
Response 2: Thank you for your advice, we have revised the title name and due to consolidating some sections together, this section is now re-classed as section 6, with the new title “Excess ROS alters mitochondrial structure, which dictates function”. Please see lines: 289, 315-316.

Round 2
Reviewer 1 Report
The reviewer appreciates any response and effort which has been done by the authors to enhance the quality of the work. After the check and correction of misspelling is carefully done again, the manuscript would be acceptable.
Reviewer 2 Report
The authors have addressed all my concerns and I have no further comments.